# Cardiometabolic health and physical robustness map onto distinct patterns of brain structure and neurotransmitter systems

Eliana Nicolaisen-Sobesky[1,2]*, Somayeh Maleki Balajoo[1,2], Mostafa Mahdipour[1,2], Agoston Mihalik[3], Mahnaz Olfati[4], Felix Hoffstaedter[1,2], Janaina Mourão-Miranda[5], Masoud Tahmasian[1,2,6], Simon B. Eickhoff[1,2], Sarah Genon[1,2]*

1 Institute of Neuroscience and Medicine (INM-7: Brain and Behaviour), Research Centre Jülich, Jülich, Germany, 2 Institute of Systems Neuroscience, Heinrich Heine University Düsseldorf, Düsseldorf, Germany, 3 Department of Psychiatry, University of Cambridge, Cambridge, United Kingdom, 4 Institute of Medical Science and Technology, Shahid Beheshti University, Tehran, Iran, 5 UCL Hawkes Institute, Department of Computer Science, University College London, London, United Kingdom, 6 Department of Nuclear Medicine, University Hospital and Medical Faculty, University of Cologne, Cologne, Germany

* elinicolaisen@gmail.com (EN-S); s.genon@fz-juelich.de (SG)

## Abstract

The link between brain health and risk/protective factors for non-communicable diseases (such as high blood pressure, high body mass index, diet, smoking, physical activity, etc.) is increasingly acknowledged. However, the specific effects that these factors have on brain health are still poorly understood, delaying their implementation in precision brain health. Here, we studied the multivariate relationships between risk factors for non-communicable diseases and brain structure, including cortical thickness (CT) and gray matter volume (GMV). Furthermore, we adopted a systems-level perspective to understand such relationships, by characterizing the cortical patterns (yielded in association to risk factors) with regards to brain morphological and functional features, as well as with neurotransmitter systems. Similarly, we related the pattern of risk/protective factors dimensions with a peripheral marker of inflammation. First, we identified latent dimensions linking a broad set of risk factors for non-communicable diseases to parcel-wise CT and GMV across the whole cortex. Data was obtained from the UK Biobank ($n = 7,370$, age range = 46–81 years). We used regularized canonical correlation analysis (RCCA) embedded in a machine learning framework. This approach allows us to capture inter-individual variability in a multivariate association and to assess the generalizability of the model. The brain patterns (captured in association with risk/protective factors) were characterized from a multilevel perspective, by performing correlations (spin tests) between them and different brain patterns of structure, function, and neurotransmitter systems. The association between the risk/protective factors pattern and C-reactive protein (CRP, a marker of inflammation) was examined using Spearman correlation. We found two significant and partly replicable latent dimensions. One latent dimension linked cardiometabolic

**Data availability statement:** Access to UKB data is explained at https://www.ukbiobank.ac.uk/enable-your-research. The code used for the machine learning framework (https://doi.org/10.5281/zenodo.7153571) has been made publicly available at https://github.com/mlnl/cca_pls_toolkit. Data supporting Figures has been shared as Supplementary Data.

**Funding:** This work was supported by the Deutsche Forschungsgemeinschaft (DFG, GE 2835/2–1, GE 2835/9-1, SFB 1451—Project-ID 431549029) (https://www.dfg.de/), granted to SG. SG received salary from GE 2835/2–1. MM received salary from GE 2835/9-1. SMB is supported by the MODS project funded from the programme "Profilbildung 2020" (grant no. PROFILNRW-2020-107-A), an initiative of the Ministry of Culture and Science of the State of Northrhine Westphalia (https://www.mkw.nrw/). SMB is supported by the EBRAINS 2.0 Project funded from the European Union's Horizon Europe Programme under the Specific Grant Agreement No. 101147319. EBRAINS is funded by the Horizon Europe Framework Programme (2023 https://ebrains.eu, https://research-and-innovation.ec.europa.eu/funding/funding-opportunities/funding-programmes-and-open-calls/horizon-europe_en). The funders did not play any role in the study design, data collection and analysis, decision to publish, or preparation of the manuscript.

**Competing interests:** The authors have declared that no competing interests exist.

**Abbreviations:** BMI, body mass index; CCA, canonical correlation analysis; CRP, C-reactive protein; CT, cortical thickness; GMV, gray matter volume; RCCA, regularized canonical correlation analysis; RSFC, resting-state functional connectivity; TIV, Total Intracranial Volume; UKB, UK Biobank.

health to brain patterns of CT and GMV and was consistent across sexes. The other latent dimension linked physical robustness (including non-fat mass and strength) to patterns of CT and GMV, with the association to GMV being consistent across sexes and the association to CT appearing only in men. The CT and GMV patterns of both latent dimensions were associated to the binding potentials of several neurotransmitter systems. Finally, the cardiometabolic health dimension was correlated to CRP, while physical robustness was only very weakly associated to it. We observed robust, multi-level and multivariate links between both cardiometabolic health and physical robustness with respect to CT, GMV, and neurotransmitter systems. Interestingly, we found that cardiometabolic health and physical robustness are associated with not only increases in CT or GMV, but also with decreases of CT or GMV in some brain regions. Our results also suggested a role for low-grade chronic inflammation in the association between cardiometabolic health and brain structural health. These findings support the relevance of adopting a holistic perspective in health, by integrating neurocognitive and physical health. Moreover, our findings contribute to the challenge to the classical conceptualization of neuropsychiatric and physical illnesses as categorical entities. In this perspective, future studies should further examine the effects of risk/protective factors on different brain regions in order to deepen our understanding of the clinical significance of such increased and decreased CT and GMV.

## Introduction

Non-communicable diseases, including cardiovascular, metabolic, mental, and neurological disorders, represent the predominant global public health challenge nowadays [1,2]. Most non-communicable diseases share a common set of risk/protective factors [3–5] (here referred simply as risk factors), like tobacco smoking, unhealthy diet, physical inactivity [1,2], excessive alcohol consumption, hypertension [2], sleep problems [6], obesity (increased body mass index (BMI)) and air pollution [2,6]. Since non-communicable diseases include mental and neurological disorders, the same set of risk factors also affects brain health [2,7].

Two of the biomarkers for brain health are cortical thickness (CT) and gray matter volume (GMV). Several studies have analyzed the link between these structural markers and specific risk factors such as BMI [8–14], waist circumference [13,14], cigarette smoking [13,15,16], physical exercise [17], and diet [18]. Even though these studies have contributed to our understanding of the link between risk factors and brain structure they have some caveats. One limitation is that the reported findings have been inconsistent across studies. For instance, associations between BMI and both, global or frontal CT, have been reported as negative [8–10,13,14], positive [11–13], or not significant [9,10]. To address this point, studies using robust approaches that evaluate the generalizability of results are needed. In this regard, machine learning approaches use cross-validation to test for the generalizability of the implemented models.

A second limitation of studies linking brain structure to risk factors is that they mostly used univariate/bivariate approaches. In other words, so far, studies have mainly linked specific risk factors with global measures of the brain, or with specific brain regions [19]. However, these factors usually do not happen in isolation (e.g., BMI, diet, and physical activity are strongly interrelated in population data) and hence represent a multivariate set. Accordingly, previous studies only provide a partial view of this association [15,16,19,20] impeding the discovery of distributed brain networks that are associated to a range of risk factors simultaneously [21]. Identifying distributed brain networks and their interactions, as well as their relationships with several risk factors requires "doubly" multivariate approaches. These approaches can take full advantage of the rich phenotyping included in large-scale datasets such as the UK Biobank (UKB) and embrace collinearity among risk factors variables, as well as among brain regions.

In this respect, canonical correlation analysis (CCA) is a multivariate, data-driven approach that can be used to discover large-scale distributed cortical patterns associated with several risk factors [22,23]. Of note, a regularized version of CCA (RCCA) mitigates the effect of collinearity and yields more stable results than CCA [22], improving the interpretability of results. CCA and RCCA have been previously used to search for robust and generalizable multivariate associations between different data modalities, such as brain structure, brain function, hippocampal structure, behavior, environmental variables, and psychiatric features [20,23–28]. Using these methods, numerous studies have shown that several healthy and illness-related phenotypes are linked to axes of brain structural organization [23,24,28].

Therefore, despite the link between risk factors for non-communicable diseases and brain health being acknowledged, there are still several important aspects of this association which are incompletely understood [21,29]. The lack of understanding of the relationship between risk factors and brain health prevents their use as biomarkers in the clinical practice and hence is a major research priority [21]. Since structural brain changes may be long-lasting and lead to various neuropsychiatric diseases, it is critical to pinpoint modifiable risk factors that can reduce the risk of those conditions. Understanding how risk factors for non-communicable diseases are related to brain health from a comprehensive and broad perspective requires understanding how multiple risk factors simultaneously relate to structural patterns across the whole brain. Thus, the main question of this study is how several risk factors for non-communicable diseases are associated with patterns of CT and GMV.

Another important point to consider is the multi-level nature of the interplay between risk factors and brain health. For instance, risk factors for non-communicable diseases have been associated with several brain features, including brain structure [30,31], function [32], genetics, and neurotransmitter systems [33]. This needed integration of different neurobiological features [34] can now be done quantitatively with neuromaps [35], which provides access to a wide set of brain maps, including, for instance, genetic transcription and brain molecular features. Similarly, a role for immunoinflammatory processes in the association between risk factors and brain health has been frequently discussed [36–40]. Linking CCA-derived cortical pattern and risk factors pattern to known neurobiological patterns and inflammation will allow to gain a systems-level understanding of the association between risk factors and brain health.

Hence, to gain a comprehensive, generalizable, and multi-level understanding of the relationship between risk factors and brain health, it is needed to use robust machine learning approaches with generalizability testing, along with methods that integrate different data modalities [28,41]. With that aim, we first searched for latent dimensions linking a wide range of risk factors for non-communicable diseases to region-wise CT and GMV across the whole brain cortex using RCCA embedded in a machine learning framework (Fig 1). In other words, we searched for those combinations of risk factors which are most relevant for CT and GMV interindividual variability [20]. On a second step, we aimed to characterize the captured brain patterns from a neurobiological perspective comparing them with existing brain maps spanning brain structure, function, genetic variability, and neurotransmitter systems. Finally, we analyzed the link between the risk factors dimension and a peripheral marker of inflammation: C-reactive protein (CRP).

PLOS Biology

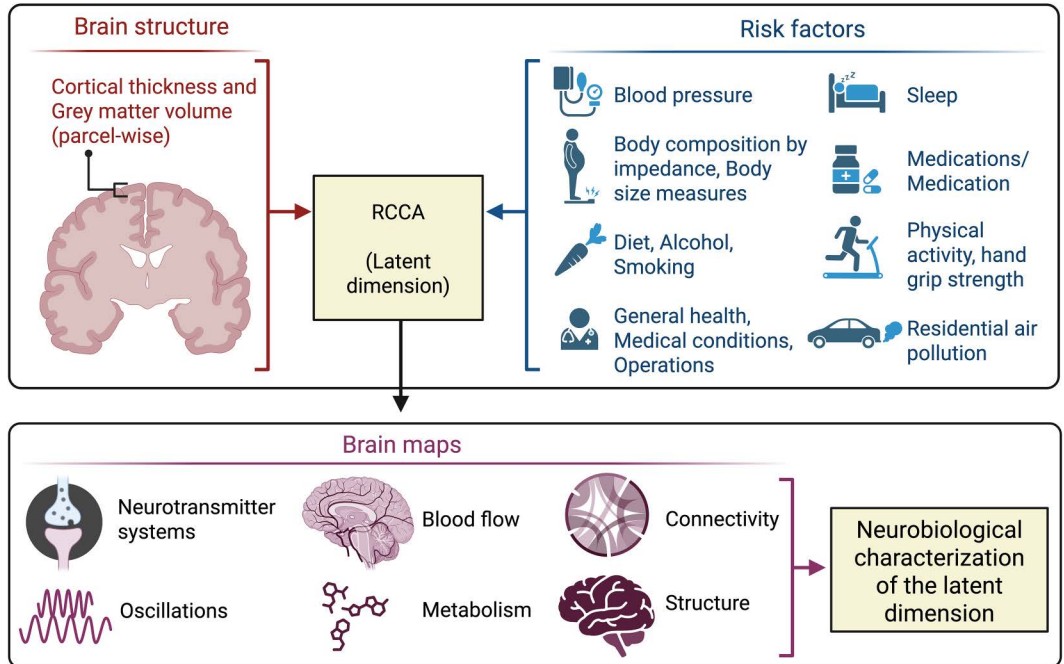

**Fig 1. Overview of the analyses.** Top panel: Latent dimensions were searched using RCCA, linking cortical thickness and gray matter volume measures (parcel-wise across the whole brain cortex) with a wide set of risk factors for non-communicable diseases. Bottom panel: In order to interpret the latent dimension, the brain loadings were compared with several brain features. Created in BioRender. Nicolaisen, E. (2025) https://BioRender.com/22pyru2.

## Methods

### Participants

We used data from UKB (application 41655) [42,43]. Inclusion criteria for this study were having no self-reported illnesses (Data-Field ID 20002-2.*) and having complete data in all variables utilized in this study (excluding responses like "Do not know" or "Prefer not to answer"). To ensure that our sample represents the male and female population equally, we selected a sample ($n = 7,370$) with an equal number of age-matched males and females. We hereafter refer to it as the main or mixed sample, since it includes individuals of both sexes/genders: 3,685 women (age: range 46–81 years, mean 62.3, standard deviation 7.6) and 3,685 men (age: range 46–81 years, mean 62.3, standard deviation 7.6). In order to check for a potential sex bias in the results, we also analyzed each sex-specific subsample, separately (see Supplementary methods 1.1 in S1 File).

### Risk factors data

We identified 68 risk factor variables in the UKB dataset (after removing variables which were duplicated (i.e., same variable measured several times), variables with missing values, and variables with skewed distribution). These 68 variables spanned categories of general health, body size measures, diet, physical activity, residential air pollution, sleep health, alcohol consumption, and smoking (Table A in S1 File). As done previously, we computed waist-to-hip ratio by dividing the waist circumference by the hip circumference [13]. All variables corresponding to risk factors were acquired on the Imaging visit, with the exception of those in the Category of "Residential air pollution" (Table A in S1 File), which were acquired before the Imaging visit.

In order to first gain a better understanding of the interrelation between risk factors alone (without taking into account brain data), the intercorrelation among risk factors was analyzed with Pearson's correlation and visualized in correlation matrices.

## Neuroimaging data

Neuroimaging data for the sample used in this study were collected by UKB in four sites using identical protocols and 3T Siemens Skyra scanners with standard Siemens 32-channel receive head coils [44]. T1-weighted structural imaging (3D-MPRAGE, sagittal) was acquired with the following parameters [42,44]: voxel resolution: $1 \times 11 \times 1$ mm, FoV: $2081 \times 2561 \times 256$ matrix, TI/TR = 880/2000 ms, in-plane acceleration iPAT = 2. T1 images were processed by UKB using a custom pipeline based on FSL [44], including gradient distortion correction, cutting down the field of view, registration (linear and then non-linear) to the MNI152 standard-space T1 template, brain extraction, defacing, and brain segmentation [44]. CT and GMV were estimated using FreeSurfer a2009s.

We used mean raw CT and mean raw GMV of 148 cortical parcels (Destrieux atlas, a2009s) [45]. To check the effect of brain size correction on the results, we also used proportional estimations of CT and GMV (computed subject-wise, dividing the mean raw CT or GMV of each parcel by the mean raw CT or Total Intracranial Volume (TIV), respectively, across all parcels) and CT and GMV corrected for brain size (regressing out mean CT or TIV, respectively, in a cross-validation consistent manner to avoid data leakage) [46]. These three different brain structural measures (raw, proportional, or corrected) represent different biological properties and can show different patterns of associations with risk factors. For instance, raw CT represents the absolute CT of a brain region, while proportional CT can be understood as the ratio between a region's CT and the whole brain's CT. Along the same line, corrected CT can be seen as the regional value adjusted for the whole brain value, that is, the regional value which is not explained by the total brain value. Accordingly, proportional and corrected values are relative values, i.e., when taking into account the whole brain value. This means that a corrected CT value for a given region expresses to which extent it has high or low CT when taking into account the global brain structure. Accordingly, corrected values are more likely to reveal structural patterns that pertain to specific brain regions. However, the drawback of correction or adjustment for head/brain size is that it may discard relevant variance, i.e., variance that pertains to the effect we are interested in. For example, if variability in brain/head size is associated with variability in physical robustness for relevant biological reasons (e.g., physically more robust individuals tend to be taller and more massive), adjusting for brain/head size can remove relevant variance related to physical robustness. Therefore, row and uncorrected/unadjusted values have pros and cons and offer different insights in the current study.

## Regularized canonical correlation analysis

CCA is a multivariate technique that can discover latent dimensions linking interindividual variability in $X$ and $Y$ [22,23,47]. Here, $X$ included either CT or GMV for each parcel, while $Y$ included risk factors. CCA searches linear combinations of variables in $X$ (brain weights $u$) and of variables in $Y$ (risk factor weights $v$), which maximize the canonical correlation between the brain scores and risk factor scores [22,23]. The scores correspond to the projection of $X$ and $Y$ onto their respective weights ($Xu$ and $Yv$) (these correspond to subject-wise values). One limitation of CCA is that it can overfit the data or yield unstable results, especially in high-dimensional datasets [48]. Therefore, we here used a regularized version of CCA (RCCA) which reduces the overfitting of the model by adding L2-norm constraints to the weights [22,23,49,50].

To search for multivariate associations between risk factors and both, CT and GMV in the main sample, we ran two RCCA models (one per brain structural measure). To check for the effect of brain size correction on the results, we ran four more analyses for cortical models: using either proportional or brain-size-corrected CT and GMV. In addition, to investigate potential sex-bias in the results, we ran 6 additional RCCA models: three in the subsample of women and three in the age-matched subsample of men (see Supplementary materials 1.1 in S1 File). Age and site effects were regressed out avoiding data leakage in the machine learning framework (regression parameters were estimated in the training set

and applied to the training, test, and holdout sets) [46,51]. In the main sample, sex was also regressed out. Confounding variables were included in the **Y** matrix to check if their variance was properly removed.

To visualize and interpret the latent dimensions, loadings were computed [22] (these correspond to variable-wise values). Brain loadings correspond to the correlation of the original brain variables (**X**) with brain scores (**Xu**). Similarly, risk factor loadings correspond to the correlation between the risk factors original variables (**Y**) with the respective scores (**Yv**). Loadings indicate which variables are more strongly associated with the latent dimension. To interpret the latent dimensions, only stable loadings were considered (loadings whose error bar did not cross zero).

### Machine learning framework

We utilized a machine learning framework that uses multiple holdouts of the data [23,52]. This framework implements two consecutive splits: the outer split divides the whole data into optimization and hold-out sets, and is used for statistical evaluation, and the inner split divides the optimization set into training and test sets and is used for model selection. In this study, we used 5 inner splits and 5 outer splits. Model selection was performed based on the highest test canonical correlation and the highest stability (similarity of weights estimated using Pearson's correlation across the 5 inner splits).

### Statistical evaluation of the latent dimensions

Statistical significance of the latent dimensions was tested with permutation tests. In each of the 1,000 iterations, the rows of the **Y** matrix were shuffled in the optimization and hold-out sets. The RCCA model (hyperparameters) that was previously selected with the original data was now fitted on the permuted optimization set, and weights were obtained. Then, the permuted hold-out set was projected onto these weights. A canonical correlation under the null model was hence obtained, and a p-value was computed. The permutation test was repeated for each one of the outer splits, hence yielding five *p*-values which were corrected by multiple comparisons (Bonferroni method over five comparisons). The statistical significance of the latent dimensions was evaluated using the omnibus hypothesis [52]. Here, the null hypothesis states that there is no effect in any of the splits. Hence, if at least one split yields a *p*-value below 0.05, the null hypothesis is rejected, and the latent dimension is considered significant. When a significant latent dimension was found, its variance was removed from the data using deflation [23], and an additional latent dimension was sought.

### Stability of the latent dimensions across sexes and brain structural measures

The latent dimensions were compared based on their average risk factor loadings and average brain loadings (average across the outer 5 splits). The risk factor loadings were compared with Pearson's correlation across models. The brain loadings were compared using a spin test with 10,000 permutations to account for spatial dependencies of brain data [53] using neuromaps software [35]. The *p*-values were corrected by multiple comparisons using Bonferroni method.

### Neurobiological characterization of the brain structural patterns

The brain maps provided in neuromaps [35] (Table C in S1 File) (excluding the map "hill2010" which is provided only in one hemisphere) were compared to the maps of CT or GMV loadings using spin test [53] with 10,000 permutations. This was performed to assess if the latent dimensions captured a CT or GMV pattern significantly associated with other patterns of brain features. Multiple comparisons were corrected using the Bonferroni method.

### Association of the latent dimensions with demographics

The association of the risk factor scores (subject-wise values) and demographics was analyzed with Spearman correlation (Supplementary materials 1.2, Table B in S1 File).

### Association of the latent dimensions with peripheral inflammation

We explore the association between the composite variable of risk factors (i.e., the risk factor scores which are subject-wise values) and a peripheral marker of inflammation with Spearman correlation. We focused on CRP (field-id 30710-1.0 in the UKB dataset) as the most relevant marker for chronic and systemic low-grade inflammation having a good signal-to-noise ratio for population studies, a relative stability in time, and a relatively low missingness in UKB. Multiple comparisons were corrected with Bonferroni.

### Latent dimensions linking risk factors to subcortical and cerebellar volumes

Although the current study focuses on cortical structure, we also added supplementary analyses examining latent dimensions yielded with subcortical and cerebellar structures for readers' information. For that, we ran one additional RCCA analysis linking the same set of risk factors as before with a set of subcortical and cerebellar volumes in the main/mixed sample (see Supplementary methods 1.3 in S1 File). Age, sex, and site were regressed out.

### Ethics

Analyses on the data have been approved by the University Hospital Düsseldorf ethics committee votes 2018-317-RetroDEuA, 2018-317_1-RetroDEuA, and 2018-317_2.

## Results

### Risk factors collinearity

The distribution of risk factors is shown in Figs A and B in S1 File. In the correlation matrices for risk factors, two groups of highly intercorrelated variables were evident (Figs C and D in S1 File). One group shows intercorrelation among body composition measures, including BMI, body fat percentage, body fat mass, body fat-free mass, body water mass, basal metabolic rate (which is estimated from fat-free body mass), impedance of whole body, waist circumference, hip circumference, and waist-to-hip ratio. Another group of highly intercorrelated variables was characterized by air pollution.

### Latent dimension linking cardiometabolic health to brain structure

A first significant latent dimension was found when analyzing the link between risk factors to raw CT in the main/mixed sample (merging women and men) ($r_{range}$ = 0.30–0.34, $p_{range}$ = 0.005–0.005) (Figs 2, E and F in S1 File, Table D in S1 File). Conceptually, this latent dimension captures variability in cardiometabolic health, for instance, with positive loadings for higher physical activity and a negative pole for factors of metabolic conditions, such as body fat, BMI, and waist-to-hip ratio, as well as blood pressure. Of note, when analyzing the link between the same set of risk factors and GMV in the main/mixed sample a latent dimension with a highly similar profile of risk factors loadings was yielded (Figs 2, E and G in S1 File, Table G in S1 File; Fig H in S1 File for comparison across latent dimensions). Note that this latent dimension corresponded to the first yielded latent dimension for all models except for the models for raw GMV, in which it was yielded as second latent dimension.

The pattern of cardiometabolic health was linked to a specific brain pattern of CT and a specific brain pattern of GMV (Fig 2). The CT pattern shows that better cardiometabolic health is associated with higher CT in the insula, cingulate cortex, temporal lobe, inferior parietal, orbitofrontal, and primary motor cortex, and to lower CT in the primary somatosensory cortex, superior frontal areas, superior parietal areas, and occipital areas. In addition, better cardiometabolic health is associated to higher GMV in primary motor cortex and temporal cortex, and lower GMV in frontal, parietal, and occipital areas.

Notably, this latent dimension of cardiometabolic health was stable across sexes and across brain size corrections (Fig H in S1 File, Supplementary results 2.1–2.3 in S1 File). Associations between the latent dimensions and demographics are shown Supplementary results 2.4 in S1 File.

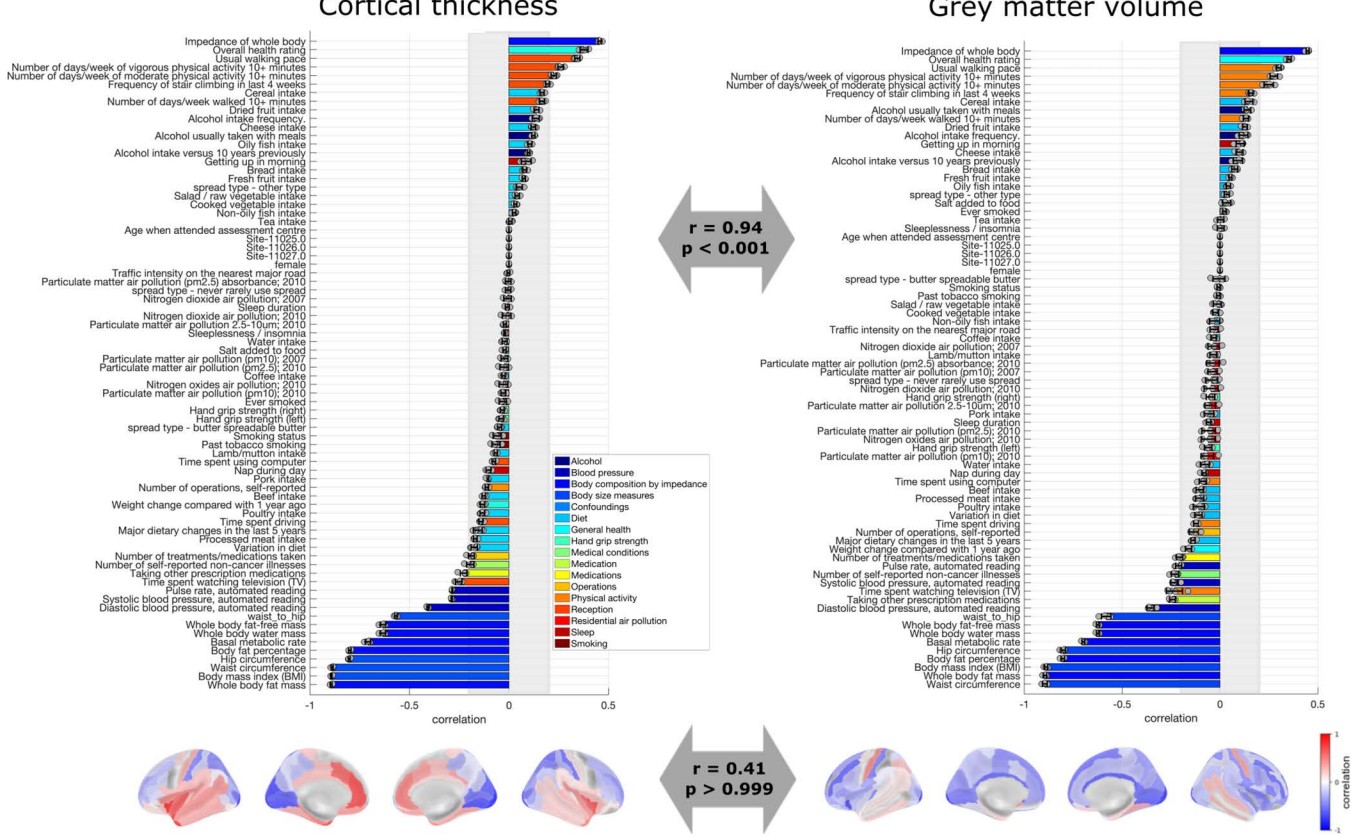

**Fig 2. Loadings of the latent dimension of cardiometabolic health.** Risk factor loadings and brain loadings in the main/mixed sample for cortical thickness (left) and gray matter volume (right). The arrows depict the Spearman correlation of the loadings across models. The loadings represent the average over the five outer splits. Error bars depict one standard deviation. The shadowed zone marks loadings between −0.2 and 0.2. The data needed to generate this figure can be found in S1 Data (left panel) and S2 Data (right panel) in S2 File.

Thus, overall, this latent dimension appears to capture a composite aspect of cardiometabolic health that robustly relates to brain structure.

### Latent dimension linking physical robustness to brain structure

Our results showed another significant latent dimension linking risk factors to raw CT ($r_{range}$ = 0.09–0.12, $p_{range}$ = 0.005–0.005) (Figs 3, Y and Z in S1 File, Table D in S1 File). Importantly, when studying raw CT, this latent dimension was significant and stable in the sample of men but not in the sample of women (see Supplementary results 2.5 in S1 File). For this reason, in the next steps, we focus on the sample of men when considering this latent dimension with respect to CT. Also, note that this latent dimension corresponded to the second yielded latent dimension for all models except for the models for raw GMV, in which it was yielded as first latent dimension. This latent dimension captured variability in physical robustness (as opposed to physical frailty), with a positive pole associated to for instance whole body fat-free mass, grip strength and overall health rating, and a negative pole associated with smoking and air pollution.

This latent dimension of physical robustness was associated with a specific brain pattern of CT and a specific brain pattern of GMV (Fig 3), though the statistical comparison showed a marginally significant Spearman correlation indicating

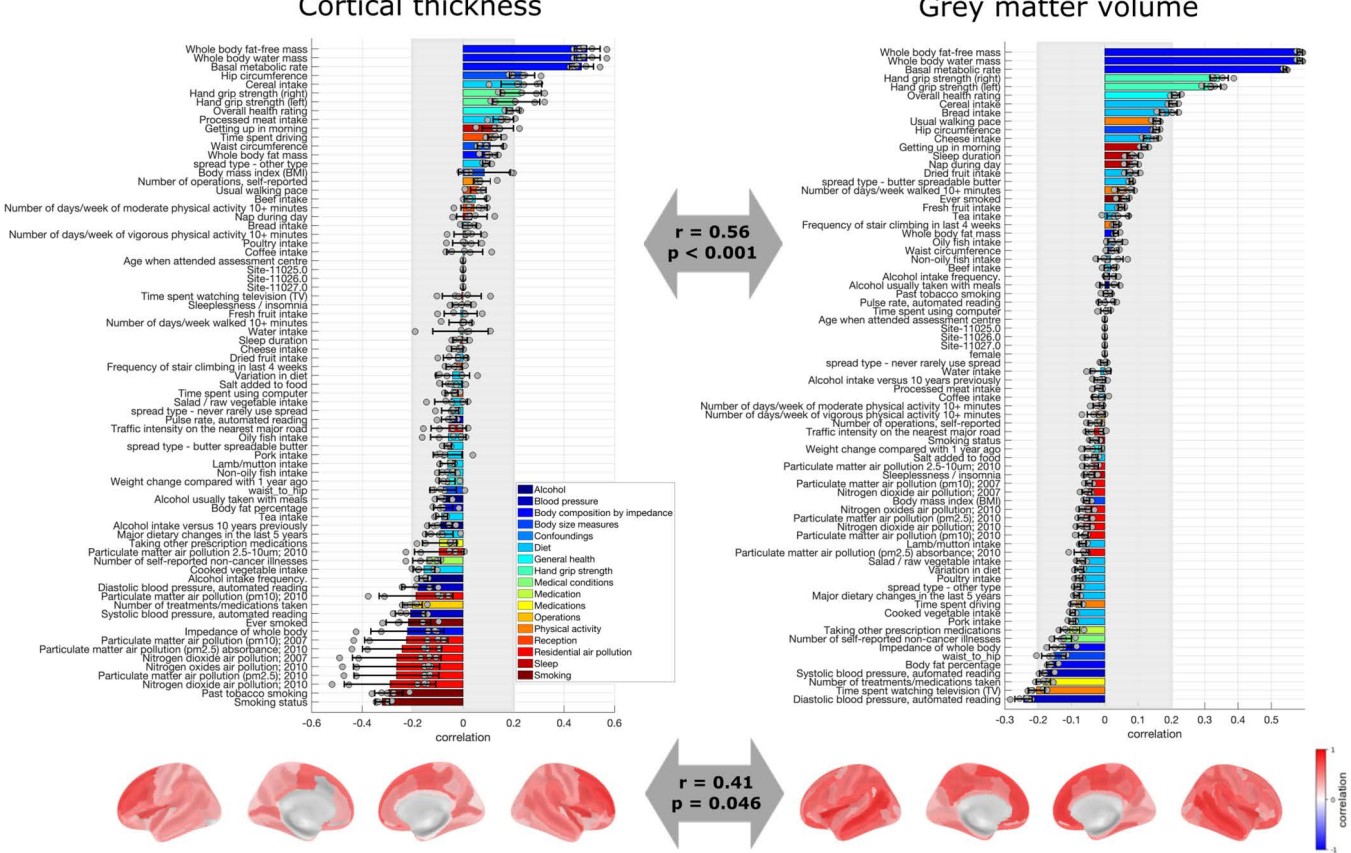

**Fig 3. Loadings of the latent dimension of physical robustness.** Risk factor loadings and brain loadings in the men sample for cortical thickness (left) and the main/mixed sample for and gray matter volume (right). The arrows depict the Spearman correlation of the loadings across models. The loadings represent the average over the five outer splits. Error bars depict one standard deviation. The shadowed zone marks loadings between −0.2 and 0.2. The data needed to generate this figure can be found in S3 Data (left panel) and S4 Data (right panel) in S2 File.

that they do share variance (*r* = 0.41, *p* = 0.046). However, this significant comparison was yielded only for the case of raw CT and raw GMV, but not for corrected brain structural measures (Fig AE-f in S1 File). Conceptually, our results indicated that higher physical robustness was associated with higher CT, especially in superior frontal areas and insula. Moreover, the physical robustness pattern was associated to higher GMV, especially in the anterior cingulate cortex, medial superior frontal areas, orbitofrontal cortex, and temporal lobe.

This latent dimension was stable across sexes when yielded with GMV, but it was not significant/stable in the sample of women when yielded with CT (Supplementary results 2.5–2.7 in S1 File). When yielded with GMV, and when yielded in the sample of men with CT, this latent dimension was stable across brain size correction procedures (Fig AE in S1 File, Supplementary results 2.5–2.7 in S1 File). Associations between the latent dimensions and demographics are shown Supplementary results 2.8 in S1 File.

### Neurobiological characterization of the pattern of brain structural loadings associated with cardiometabolic health

To characterize the latent dimensions from a neurobiological perspective, we compared the CT and GMV loadings with brain maps spanning brain function, structure, and neurotransmitter systems.

For the latent dimension associated to cardiometabolic health, several associations between the CT and GMV patterns and different brain maps were significant (Table J in S1 File, Figs 4 and AO in S1 File). Namely, the map of CT loadings was positively associated with the cortical distribution of serotonin receptor 5-HT1a [54,55], dopamine transporter DAT [56], receptor GABAa [57], and acetylcholine transporter VAChT [34,58]. In addition, the CT pattern was positively associated with CT [59], and the fifth gradient of resting-state functional connectivity (RSFC) [60], as well as negatively associated with glucose metabolism [61], histone deacetylase [62], and with the first principal component of genes in the Allen

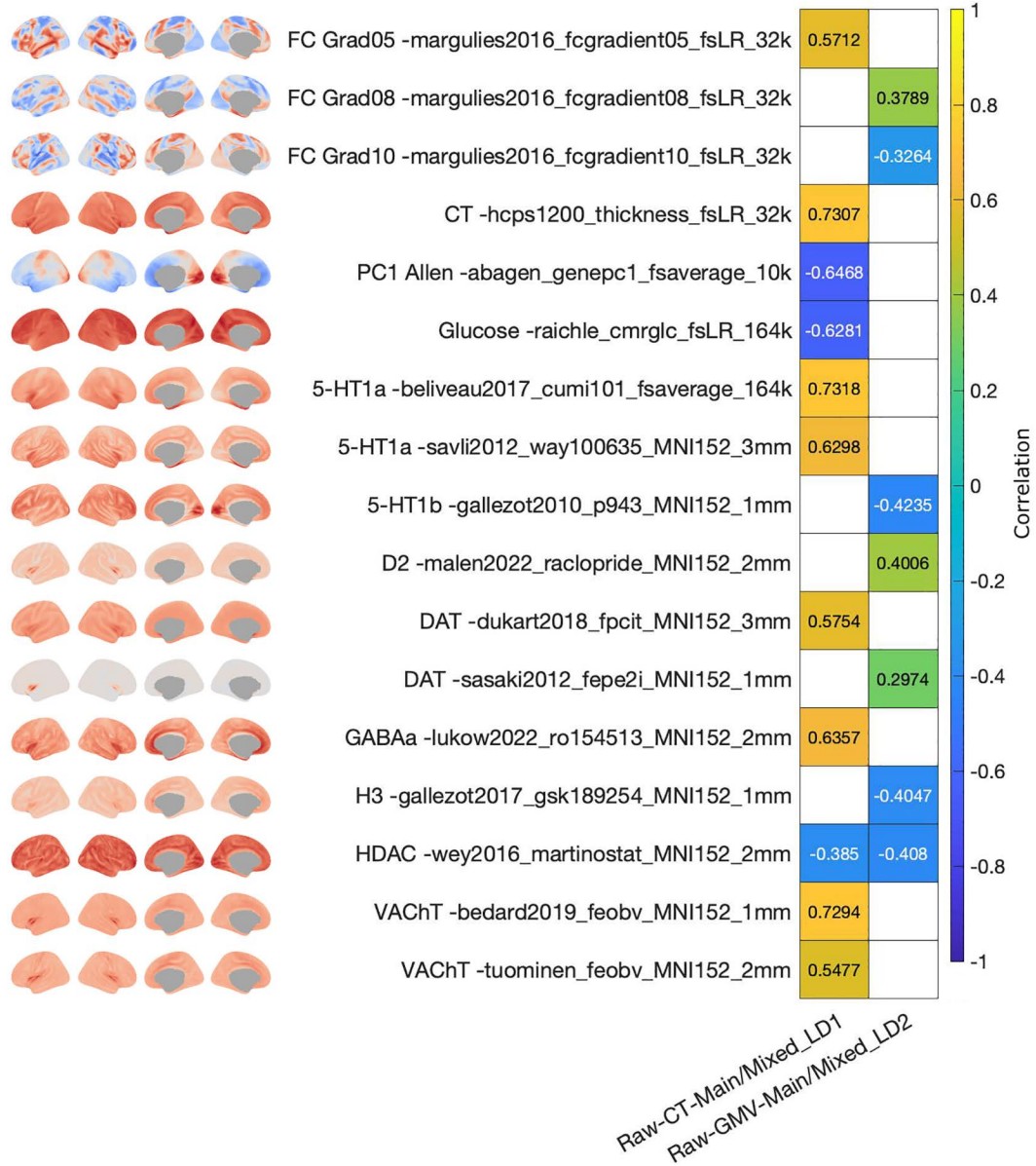

**Fig 4. Association of brain structural loadings with neuromaps for the latent dimension of cardiometabolic health.** Only data for neuromaps that yielded a significant association with at least one of the shown loadings maps are depicted. Colorbar represents the Spearman correlation. White tiles represent non-significant associations. The data needed to generate this figure can be found in S5 Data in S2 File.

Human Brain Atlas [63,64]. In turn, the GMV pattern of the cardiometabolic health latent dimension was positively associated with the cortical distribution of dopamine receptor D2 [65] and dopamine transporter DAT [66], and negatively associated with serotonin receptor 5-HT1b [34,67], and histamine receptor H3 [68]. Moreover, the GMV pattern was positively associated with the 8th gradient of RSFC [60], and negatively associated with histone deacetylase [62] and 10th gradient of RSFC [60].

Conceptually, this indicates that the latent dimension captured a brain axis in which cortical regions whose CT covaried positively the most (increased the most, red regions in Fig 2) with the pattern of cardiometabolic health were regions that showed the highest binding potentials for 5-HT1a, DAT, GABAa, and VAChT. Similarly, the brain regions whose GMV increased the most (red regions) with cardiometabolic health were regions with the highest binding potential for D2 and DAT. In turn, the brain regions whose GMV covaried negatively the most (decreased the most, blue regions) with cardiometabolic health were regions with the highest binding potentials for 5-HT1b and H3.

### Neurobiological characterization of the pattern of brain structural loadings associated with physical robustness

The brain patterns yielded for the latent dimension of physical robustness also showed several significant associations with brain maps (Table K in S1 File, Figs 5 and AP in S1 File). The CT pattern yielded in the sample of men was positively associated with glutamate receptor mGluR5 [34,69], RSFC intersubject variability [70], and developmental area scaling [71]. The GMV pattern was positively associated with the serotonin receptors 5-HT1a [54], with glutamate receptors mGluR5 [34,69,72] and NMDA [34,73–75], with histamine receptor H3 [68], with opioid receptors MOR [76,77] and KOR [78], with cannabinoid receptor CB1 [79,80], and with receptor GABAa [57]. In addition, the GMV pattern was positively associated with the 1st gradient of RSFC [60], intersubject variability of RSFC [70], CT [59], developmental area scaling [71], evolutionary expansion [81], and sensory–association axis [82]. Moreover, the GMV pattern was negatively associated with T1w/T2w [59] and with functional homology [81].

Conceptually, this indicates that the latent dimension captured a brain axis in which cortical regions whose CT increased the most with the pattern of physical robustness (red regions in Fig 3) were those with highest binding potential for mGluR5. In addition, regions whose GMV increased the most with physical robustness (red regions) were those with the highest binding potential for 5-HT1a, mGluR5, NMDA, H3, MOR, KOR, CB1, and GABAa.

### Correlation of the latent dimensions with peripheral marker of inflammation

CRP was significantly associated to the risk factors scores of cardiometabolic health when yielded in association with raw CT ($r=-0.39$, $p<0.001$) and with raw GMV ($r=-0.39$, $p<0.001$) in the main/mixed sample. It was also significantly associated to the risk factors pattern of physical robustness yielded in association with raw GMV in the main/mixed sample ($r=-0.08$, $p=0.04$) but with a very low effect size. CRP was not significantly associated with the risk factors yielded in association with raw CT in the sample of men ($r=0.038$, $p=0.999$). Thus, overall, the composite variable of cardiometabolic health appears moderately correlated to a marker of low-grade systemic inflammation, while physical robustness was only very weakly associated ($r<0.1$) to such a marker.

### Latent dimensions linking risk factors to subcortical and cerebellar volumes

Results yielded by the analysis of subcortical and cerebellar data are reported in Supplementary results section 2.9 in S1 File. Overall, more significant latent dimensions appeared, and only one was partly related to the two main dimensions or risk factors found in association to cortical structure. This suggests that subcortical regions have different patterns of interindividual variability than cortical regions (likely higher interindividual variability and more vulnerability to exposome factors). Accordingly, their associations with risk factors should be the focus of specific studies. In the following discussion, we focus on cortical structure.

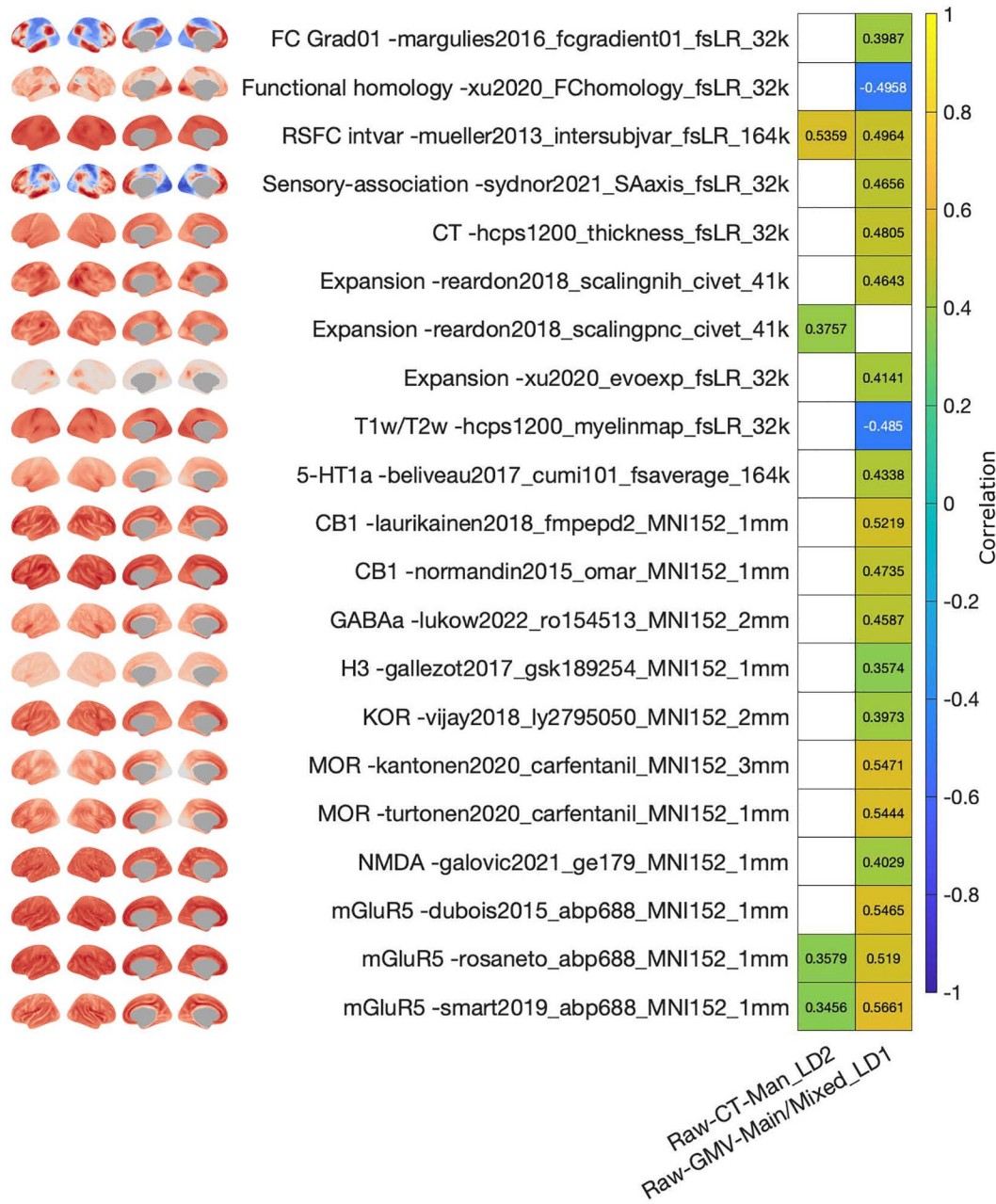

**Fig 5. Association of brain structural loadings with neuromaps for the latent dimension of physical robustness.** Only data for neuromaps that yielded a significant association with at least one of the shown loadings maps are depicted. Colorbar represents the Spearman correlation. White tiles represent non-significant associations. The data needed to generate this figure can be found in S6 Data in S2 File.

## Discussion

We report two latent dimensions characterizing the interplay between a wide range of risk factors for non-communicable diseases with region-wise CT and GMV across the whole cortex. One of these latent dimensions highlighted the relevance of cardiometabolic health for inter-individual variability of brain structure in a healthy sample. Importantly, this latent

dimension was stable across sexes, across brain structural measures (CT and GMV), and across brain size corrections (raw, proportional, and brain-size corrected CT). Accordingly, this latent dimension cannot be explained by a confounding effect of head size/morphology that could be conveyed in brain structural estimates. The other latent dimension highlights the relationship between physical robustness and brain structure. Even though this latent dimension was stable within analyses, the stability across sexes, brain structural measures, and brain size corrections was limited. In addition, our results underline the multi-level nature of the association between risk factors and brain structure, linking the brain patterns of both latent dimensions with the spatial distribution of several neurotransmitter systems.

We showed that the variability in cardiometabolic health is related to an axis of variability in CT extending from the insula and cingulate cortex to the occipital lobes and superior parietal regions. Overall, this latent dimension indicates that regions engaged in processing internal information such as emotional and motivational systems (including, for instance, insula, orbitofrontal cortex, and anterior cingulate cortex) show positive loadings. Accordingly, this CT pattern was significantly related to dopamine and serotonin systems. In contrast, more dorsal regions typically engaged in dorsal attention and executive systems (such as the lateral superior prefrontal and parietal cortex) show negative CT loadings on the latent dimension. Similarly, cardiometabolic health was negatively associated with GMV in the frontal regions. However, it was positively associated with GMV primarily in sensorimotor regions (peri-central regions), but also to a lesser extent to the ventral attention network (lateral parietal and middle temporal regions). This suggests that cardiometabolic health may be differently related to variability in CT and variability in GMV in the population. More concretely, our results suggest that better cardiometabolic health is related to increased CT in emotional and motivational systems along with increased GMV in sensorimotor cortex. Future studies should further elucidate the mechanism behind these associations to identify whether improving cardiometabolic health can promote brain structural health in motivation, emotion, and sensorimotor networks and hence improve mental health and motor function in aging. The potential role of neurotransmitters system in these mechanisms is further discussed below.

In the current stage of knowledge, our results are in line with recent reports pointing to the important role of cardiometabolic health for brain health. For instance, several studies have linked cardiometabolic factors to brain structure, such as reduced total brain volumes [38,83,84], reduced GMVs [31,38,84], or to structural markers of brain aging [85]. Interestingly, our findings are in contrast with works that have reported only reductions, but not increases, in brain structural measures in association with risk factors or cardiometabolic health. Differences in the health status of the samples might explain these differential effects. Apart from associations with brain structure, cardiometabolic health has also been associated with brain function, such as cognition [38], dementia [83], and other neuropsychiatric disorders and symptoms [21,29,38,86,87]. In our study, a moderate correlation between the composite variable of cardiometabolic health and a marker of peripheral inflammation further suggests a role for low-grade chronic inflammation in the association between cardiometabolic health and brain health.

Actually, several studies have pointed out that a causal factor or mediator in the association between risk factors for non-communicable diseases and brain health might be inflammation [36–40], and in particular, low-grade systemic inflammation [21,36,38,39,88,89]. Risk factors for non-communicable diseases are also risk factors for low-grade systemic chronic inflammation [6,39,89]. In turn, low-grade systemic chronic inflammation has been associated with non-communicable diseases [6,88], including neuropsychiatric illnesses [21,38,39,88–92]. For instance, chronic inflammation in the adipose tissue has been associated with the development of neurodegenerative disorders such as Alzheimer's disease [39,93]. Moreover, inflammatory factors have been reported to mediate the association between BMI and other risk factors with cortical structure and behavior [36,37] (however, see [94]). In fact, inflammation has been proposed as the cause of comorbidities between non-communicable diseases [86,90], such as depression and cardiovascular or neurodegenerative illnesses [90]. Thus, we can speculate here that cardiometabolic health, partly via inflammation pathways, influences brain health with potential impacts on emotion and motivation behavioral systems, as well as to motor systems.

Our results also show that interindividual variability in physical robustness (fat-free mass and strength) is associated with interindividual variability in brain structure, in both men and women for GMV, but only robustly in men for CT. At the brain level, this CT pattern in men shows the highest loadings mainly across the whole lateral frontal cortex and the superior medial frontal gyrus. The pattern of GMV associated to physical robustness also shows high loadings on medial frontal regions, along with lateral temporal regions. Accordingly, this GMV pattern is also significantly related to cortical patterns of developmental area scaling, along with evolutionary expansion and sensory–association axis. This may suggest common mechanisms underlying physical health (likely musculoskeletal health) and brain health in key regions for human high-level cognition (such as the frontal cortex). Interestingly, time spent watching television appears on the negative pole of these dimensions suggesting that sedentary lifestyles may threaten this aspect of human brain health.

It should be noted that the variability of physical robustness in women appears to be linked to GMV but not to CT, while in men it is associated to both structural markers. This sex-variability could be explained by the risk factors profile in association with CT in men capturing a negative pole of air pollution and smoking. Previous studies have pointed towards sex-variability in the effect of air pollution on health and particularly on brain and behavior [95–99]. This could be due to sex-specific exposures to pollutants, with more interindividual variability in exposures in men allowing us to observe associations with health outcomes. In line with this hypothesis, the distribution of variables related to air pollution in our sample of men includes a few more extreme values than in women (Fig B in S1 File). Alternatively, a potential sex-specific effect of air pollutants could also be due to different biological aspects being differentially affected in men in comparison to women, or to sex-specific pathways underpinning the association between air pollution and brain structure. The mechanisms underpinning this difference should be explored in future studies. Moreover, studies analyzing the association between health variables and neuroimaging should consider this and perform sex-specific analyses. In the current stage of knowledge, we can only speculate from our results that air pollution and smoking may represent a specific threat for men brain health in high-level association regions.

Interestingly, the brain patterns associated to both dimensions of risk factors were generally significantly correlated to the spatial distribution of most neurotransmitter systems. It should also be noted that the neurotransmitters associated to the latent dimension have in turn been linked to phenotypes related to the risk factors captured in the latent dimension. For instance, the serotoninergic system is associated with energy balance and feeding behavior [100,101], obesity [100], and physical activity [102]. Specifically, the 5-HT1a receptor has been associated with food intake [101], anorexia nervosa, and bulimia nervosa [103]. The dopaminergic system has been associated with physical activity [102], and specifically, the receptor D2 has been associated with disorders related to eating behavior, such as anorexia nervosa, bulimia nervosa, and obesity [103]. Overall, the evidence indicates that these neurotransmitter systems are associated with imbalances in energy homeostasis and feeding behavior, which are phenotypes related to the pattern of risk factors captured in our latent dimension. This suggests that the clinically relevant interaction between brain and body health that has been called to attention recently [21,29] might be mediated by processes associated with neurotransmitter systems and/or importantly alter neurotransmitter systems. For instance, cardiometabolic factors may lead to perturbations in the serotoninergic system via brain structural alterations, ultimately leading to immune-metabolic depression [104].

Evidence of the mechanistic cause linking neurotransmitter systems with both, risk factors for non-communicable diseases and brain structure, also points to inflammation. Several studies have shown a crosstalk between the immune system and several neurotransmitter systems, such as serotoninergic, noradrenergic, and dopaminergic systems [92,105]. For instance, certain neurotransmitter receptors, including 5-HT1a, D1, and D2, have immunologic functions [88,105], and can lead to the disruption of homeostasis [88] and to inflammation. Accordingly, immune cells express neurotransmitter receptors [105]. In turn, immunological factors can regulate normal cellular functions, including neurotransmission and synaptic plasticity [88]. In sum, the crosstalk between inflammatory factors and neurotransmitter systems is bidirectional [88,105] and is relevant for several non-communicable diseases [105]. Moreover, since these neurotransmitter systems are implicated in mental disorders and in somatic non-communicable diseases [88,90], the mechanisms underlying the

comorbidity between mental and somatic disorders might be partly related with alterations in neurotransmitter systems [86]. Accordingly, it is likely that a vicious cycle is engaged when cardiometabolic and musculoskeletal systems, immunoinflammatory systems, brain structure, and neurotransmitters systems are altered complexifying treatment of neuropsychiatric disorders.

In that context, our findings contribute to the recognized urgency to characterize brain-body interactions for its implementation in clinical practice [21,29,86]. For instance, it is not common to monitor physical illnesses in patients with neuropsychiatric disorders. However, recently it has been pointed out that neuropsychiatric disorders are associated with symptoms of physical illnesses and that poor physical health is a more pronounced effect than brain phenotypes in these patients [29]. Given that the interplay between brain and body health is not well understood, the clinical practice nowadays has limited tools to exploit this interaction, not only for a comprehensive monitoring of health, but also for its use as biomarkers. Hence, research characterizing brain and body interactions is a major priority for global health because it will guide the discovery of new integrated disease manifestations and pave the way for the development of new therapies and clinical interventions [21,29].

## Conclusions

Our study shows two latent dimensions linking cardiometabolic health and physical robustness to patterns of CT and GMV variability across the whole cortex. In turn, the captured CT and GMV brain patterns were associated with the cortical distribution of several neurotransmitter systems. Furthermore, cardiometabolic health was associated to a peripheral marker of low-grade systemic inflammation. Hence, our study shows that body health, brain structure, and neurotransmitter systems are interrelated, highlighting the multi-level nature of health. Also, our work contributes to questioning the classic consideration of neuropsychiatric and somatic illnesses as separate categories [88] and supports the view of a needed integration of brain and physical health in clinical practice [21,29].

## Supporting information

**S1 File. Including: Supplementary methods, Supplementary results, Supplementary tables, Supplementary figures.**
(PDF)

**S2 File. Including: S1 Data–S75 Data.**
(ZIP)

## Acknowledgments

The contribution of ENS has been done in partial fulfillment of the requirements for a Ph.D. thesis. This research has been conducted using the UK Biobank Resource under Application Number 41655.

## Author contributions

**Conceptualization:** Somayeh Maleki Balajoo, Masoud Tahmasian, Sarah Genon.

**Data curation:** Eliana Nicolaisen-Sobesky, Somayeh Maleki Balajoo, Mostafa Mahdipour, Felix Hoffstaedter, Sarah Genon.

**Formal analysis:** Eliana Nicolaisen-Sobesky, Sarah Genon.

**Funding acquisition:** Simon B. Eickhoff, Sarah Genon.

**Investigation:** Eliana Nicolaisen-Sobesky, Mostafa Mahdipour, Felix Hoffstaedter, Masoud Tahmasian, Simon B. Eickhoff, Sarah Genon.

**Methodology:** Eliana Nicolaisen-Sobesky, Somayeh Maleki Balajoo, Agoston Mihalik, Mahnaz Olfati, Janaina Mourão-Miranda, Masoud Tahmasian, Sarah Genon.

**Project administration:** Sarah Genon.

**Resources:** Simon B. Eickhoff, Sarah Genon.

**Software:** Eliana Nicolaisen-Sobesky, Agoston Mihalik, Janaina Mourão-Miranda.

**Supervision:** Somayeh Maleki Balajoo, Sarah Genon.

**Visualization:** Eliana Nicolaisen-Sobesky.

**Writing – original draft:** Eliana Nicolaisen-Sobesky, Sarah Genon.

**Writing – review & editing:** Eliana Nicolaisen-Sobesky, Somayeh Maleki Balajoo, Mostafa Mahdipour, Agoston Mihalik, Mahnaz Olfati, Felix Hoffstaedter, Janaina Mourão-Miranda, Masoud Tahmasian, Simon B. Eickhoff, Sarah Genon.

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
