## [Editor Report · Decision Letter 0]

19 Feb 2025

Dear Dr Nicolaisen-Sobesky,

Thank you for submitting your manuscript entitled "Cardiometabolic health, cortical thickness, and neurotransmitter systems: a large-scale multivariate study" for consideration as a Research Article by PLOS Biology and please accept our apologies for the delay in sending you an initial decision. We had wished to discuss your paper with an Academic Editor who works in this field, and it took us a bit longer than normal to find someone who was available to provide advice.

Your manuscript has now been evaluated by the PLOS Biology editorial staff and by an academic editor with relevant expertise and I am writing to let you know that we would like to send your submission out for external peer review.

Once your full submission is complete, your paper will undergo a series of checks in preparation for peer review. After your manuscript has passed the checks it will be sent out for review. To provide the metadata for your submission, please Login to Editorial Manager (https://www.editorialmanager.com/pbiology) within two working days, i.e. by Feb 21 2025 11:59PM.

Kind regards,

Luke

Lucas Smith, Ph.D.

Senior Editor

PLOS Biology

lsmith@plos.org

---

## [Decision Letter · Decision Letter 1]

9 Apr 2025

Dear Dr Nicolaisen-Sobesky,

Thank you for your patience while your manuscript "Cardiometabolic health, cortical thickness, and neurotransmitter systems: a large-scale multivariate study" was peer-reviewed at PLOS Biology. It has now been evaluated by the PLOS Biology editors, an Academic Editor with relevant expertise, and by several independent reviewers.

In light of the reviews, which you will find at the end of this email, we would like to invite you to revise the work to thoroughly address the reviewers' reports.

As you will see below, the reviewers have commented that the study offers some interesting insights, but they have also raised a number of concerns and provided suggestions to strengthen the study further and we think the reviewer comments will need to be thoroughly addressed before we can consider the study for publication. We think the revised study will need to be developed further, with new analyses to interrogate the sex differences in more detail, as reviewer 1 suggests, and to look at subcortical structures and add cortical area and/or volume as reviewer 2 suggests.

Given the extent of revision needed, we cannot make a decision about publication until we have seen the revised manuscript and your response to the reviewers' comments. Your revised manuscript is likely to be sent for further evaluation by all or a subset of the reviewers.

We expect to receive your revised manuscript within 3 months, however if you need an extension to complete these revisions please let me know. I would be happy to extend the deadline for the revision by a couple of months, if needed. Please email us (plosbiology@plos.org) if you have any questions or concerns, or would like to request an extension.

**IMPORTANT - SUBMITTING YOUR REVISION**

*Re-submission Checklist*

*Published Peer Review*

*PLOS Data Policy*

*Blot and Gel Data Policy*

Sincerely,

Luke

Lucas Smith, Ph.D.

Senior Editor

PLOS Biology

lsmith@plos.org

REVIEWS:

Reviewer #1: The research presented uses regularized canonical correlation analysis (RRCA) to investigate relationships between factors related to cardiometabolic health and cortical thickness (CT). The weightings for significant imaging latent dimensions were then compared to atlases relating to neurotransmitters, brain metabolism, genetics and brain structure and function. Analyses were repeated for males, females, and all participants using raw, proportional, and corrected CT.

The authors concentrate on the first latent dimension, which they link to "cardiometabolic health". This factor has the same sign of loadings for body fat, body water, and fat-free mass. Interestingly, the second latent dimension (Figure S9) shows opposite signs for fat-free and water mass compared to body fat percentage, although this dimension seems to be driven more by men (Figure S20) than by women (Figure S19).

The authors state that "the second and third latent dimensions were also significant. However, the results point towards less cross-sex shared variability for the second and third latent dimensions given that they were unstable in the sex-specific subsamples".

It seems that in analyzing men and women separately, there is no guarantee of correspondence between the same latent dimension in the two groups. It appears that men show a much more robust second latent dimension (Figure S20) than women (Figure S19). Men also show a fairly robust third latent dimension based on residential air pollution and smoking (Figure S21), although I don't see the corresponding data for women. There is an existing literature suggesting differential effects of air pollution on males and females which may or may not support the findings reported.

Rather than dismissing this cross-sex variability, I would encourage the authors to embrace these interesting findings. This could be done by either calculating the loadings based on both groups, and then reporting the corresponding correlations for men and women, or by reporting and discussing separate RCCA of each group.

Minor comments

Loadings Figures. Given that there only 68 risk factors available, it would seem more consistent to include all loadings in each diagram, even if they are not "stable". The brain loadings are important to the interpretation of the results. It may be better to include only medial and lateral views, but to make them larger.

It would be useful to be able to visually assess the associations between cortical thickness loadings and neuromaps (Figure 3), maybe with representative brain images side-by-side.

The p-values reported in the main paper and the supplementary materials appear to be quantized in steps of 0.005, e.g. "rrange=0.30-0.34, p=0.005-0.005", Tables S4-S12.

The ordering of risk factors in Figures S3 and S4 could be changed to better reflect the domains so that, for example, BMI and waist circumference are close together.

The authors suggest in the abstract that "regular monitoring of cardiometabolic risk factors should be considered in healthcare practice". While I don't disagree with this statement, it is already clinically well established that risk factors such as smoking, obesity, and hypertension cause neurovascular disease (often visible on routine MRI) that will almost certainly impact brain health (including stroke risk), and presumably cortical thickness.

Table 1. Given a consistent number of regions, I would expect a monotonic relationship between the correlation coefficient and p-value. The numbers in this table look odd, for example row 16 has r=0.55, p=0.26 but row 19 has r=0.55, p<0.001. Again, there appears some quantization of p-vales (e.g. p=0.26 appears four times in the first column).

Reviewer #2: This study revealed a latent pattern of brain cortical thickness that associated with a range of cardiometabolic risk factors in a group of healthy individuals participating in the UK Biobank. They further showed that the latent cortical pattern is associated with the spatial distribution of several neurotransmitters. This is an interesting study, and the statistical analysis is robust.

* My main concern is the lack of interpretation of the revealed CT pattern and their associated cardiometabolic risk factors. Although some interpretation is in place, it is insufficient to understand why increased and decreased CT differentially associated cardiometabolic risk factors across regions.

* Another concern relates to the choice of the three different but related CT measures. The reproducible findings across the three measures are not surprising given that they are correlated. It would be helpful to elaborate the rationale of selecting the three measures and clarify the "different biological properties" mentioned in the manuscript.

* Similarly, the rationale of choosing CT over volume, surface area or other metrics are unclear. Gray matter volume is also an important marker of brain health.

* The choice of focusing on the cerebral cortex and ignoring the subcortical structure is also unclear. Subcortical nuclei/regions have been widely implicated in brain disorders (e.g., hippocampus in dementia, striatum in psychosis etc) that often comorbid with cardiometabolic conditions. It would be important to consider the subcortex in my opinion.

Reviewer #3, Yunpeng Wang (note, Reviewer 3 has signed this review): Review of Nicolaisen-Sobesky et al. for PLOS Biology

Nicolaisen-Sobesky et al. investigated the relationship between 68 cardiometabolic variables and brain cortical thickness using a carefully selected subset of the UK Biobank dataset. Unlike previous work, this study employed regularized canonical correlation analysis (RCCA), which accounts for dependencies within both sets of variables. This approach allows for a more holistic understanding of brain-body relationships. The authors identified several statistically significant associations, which appear robust and generalizable within UKBB. They further linked the latent dimensions to brain maps derived from external datasets.

The novelty of this study includes:

An important contribution to the literature on brain-body connections, particularly in the context of commonly studied physical health risk factors and MRI-derived brain features.

The finding that some risk factors are associated with increased cortical thickness in certain regions and decreased thickness in others. This challenges the assumption—common in prior literature—that thicker cortex uniformly implies better function.

Integration of latent dimensions with previously established brain maps.

However, several issues need to be addressed:

Minor:

Some references need attention. For instance:

References 22 and 27 appear to be duplicates.

Reference 38 lacks complete journal information.

Reference 75 may not be the correct citation.

Major:

Repeated measures: It is unclear which measurements of the risk factors were used. Did the authors use the measurement closest in time to the brain scan? Were other criteria used? This should be clearly described.

Overemphasis on inflammation in the Discussion: About 60% of the Discussion focuses on inflammation and chronic inflammation as potential mechanisms. While this is a plausible interpretation, the UKBB includes actual inflammatory biomarkers (e.g., CRP, OLINK inflammation panel). The authors should consider testing these directly or, at the very least, connect their current findings to previous results based on these measures. For example, are there risk factors that showed strong associations in univariate analyses but not here, or vice versa?

In sum, while the methodological approach and findings are interesting and potentially impactful, the manuscript would benefit from clearer methodological details and a stronger integration with prior work.

---

## [Decision Letter · Decision Letter 2]

2 Oct 2025

Dear Dr Nicolaisen-Sobesky,

Thank you for your patience while we considered your revised manuscript "Risk factors, brain structure, and neurotransmitter systems: a large-scale multivariate study" for publication as a Research Article at PLOS Biology. This revised version of your manuscript has been evaluated by the PLOS Biology editors, the Academic Editor, and the original reviewers.

Based on the reviews, we are likely to accept this manuscript for publication, provided you satisfactorily address the following data and other policy-related requests.

IMPORTANT - please attend to the following:

a) PLOS Biology Titles should contain an active verb and avoid unnecessary punctuation. Please change your Title to something like: "A large-scale multivariate study reveals associations between disease risk factors, brain structure and neurotransmitter systems"

b) Please address my Data Policy requests below; specifically, we need you to supply the numerical values underlying Figs 2, 3, 4, 5, S1-S48, either as a supplementary data file or as a permanent DOI’d deposition. I note that you currently mention the constraint placed on the raw individual-level data in UK BioBank, but what we’re after is the processed numerical values that directly underlie the Figures. You can see examples of the data availability statements and provisions of other PLOS Biology papers that make use of UK BioBank data here: https://journals.plos.org/plosbiology/article?id=10.1371/journal.pbio.3001768 and https://journals.plos.org/plosbiology/article?id=10.1371/journal.pbio.3001656

c) Please cite the location of the data clearly in all relevant main and supplementary Figure legends, e.g. “The data underlying this Figure can be found in S1 Data” or “The data underlying this Figure can be found in https://zenodo.org/records/XXXXXXXX

d) Please include the entire “Data and code sharing statement” (currently in your manuscript) in the Data Availability statement in the Editorial Manager metadata (i.e. including the info about Zenodo and Github).

We expect to receive your revised manuscript within two weeks.

*Published Peer Review History*

*Press*

Sincerely,

Roli Roberts

Roland G Roberts PhD

Senior Editor

rroberts@plos.org

PLOS Biology

on behalf of

Lucas Smith, Ph.D.

Senior Editor

lsmith@plos.org

PLOS Biology

DATA POLICY:

Regardless of the method selected, please ensure that you provide the individual numerical values that underlie the summary data displayed in the following figure panels as they are essential for readers to assess your analysis and to reproduce it: Figs 2, 3, 4, 5, S1-S48. NOTE: the numerical data provided should include all replicates AND the way in which the plotted mean and errors were derived (it should not present only the mean/average values).

CODE POLICY

DATA NOT SHOWN?

REVIEWERS' COMMENTS:

Reviewer #1:

I thank the authors for their careful consideration and thoughtful responses to the reviewers' comments and suggestions. The manuscript now addresses all the points raised in review, and I am happy to recommend acceptance for publication.

Reviewer #2:

[identifies herself as Ye Ella Tian]

All my previous comments have been appropriately addressed. I do not have further comments.

Reviewer #3:

[identifies himself as Yunpeng Wang]

Thank you for the efforts. I am satisfied with your responses. This is a great paper!

---

## [Editor Report · Decision Letter 3]

28 Oct 2025

Dear Dr Nicolaisen-Sobesky,

Thank you for the submission of your revised Research Article "Cardiometabolic health and physical robustness map onto distinct patterns of brain structure and neurotransmitter systems" for publication in PLOS Biology, and thank you for addressing our last editorial requests in this revision. On behalf of my colleagues and the Academic Editor, Marcus Munafò, I am pleased to say that we can in principle accept your manuscript for publication, provided you address any remaining formatting and reporting issues. These will be detailed in an email you should receive within 2-3 business days from our colleagues in the journal operations team; no action is required from you until then. Please note that we will not be able to formally accept your manuscript and schedule it for publication until you have completed any requested changes.

PRESS

Sincerely, 

Luke

Lucas Smith, Ph.D.

Senior Editor

PLOS Biology

lsmith@plos.org